# Problematic Use of Smartphones and Social Media on Sleep Quality of High School Students in Mexico City

**DOI:** 10.3390/ijerph21091177

**Published:** 2024-09-04

**Authors:** Cristopher Martín Olivares-Guido, Silvia Aracely Tafoya, Mónica Beatriz Aburto-Arciniega, Benjamín Guerrero-López, Claudia Diaz-Olavarrieta

**Affiliations:** 1Department of Psychiatry and Mental Health, Faculty of Medicine, National Autonomous University of Mexico, 3000 Ave. Universidad, Ciudad Universitaria, Coyoacán, Mexico City 04510, Mexico; cristopher.olivares@cch.unam.mx (C.M.O.-G.); stafoya@unam.mx (S.A.T.); dr.bguerrero@unam.mx (B.G.-L.); 2Research Division, Faculty of Medicine, National Autonomous University of Mexico, 3000 Ave. Universidad, Ciudad Universitaria, Coyoacán, Mexico City 04510, Mexico; maburto@unam.mx

**Keywords:** problematic internet use, problematic mobile phone use, social media, sleep quality, high school students

## Abstract

Background: Smartphones, internet access, and social media represent a new form of problematic behavior and can affect how teens sleep. Methods: A cross-sectional design was employed to examine the prevalence and association of problematic internet use and problematic smartphone use with sleep quality in a non-probability sample of 190 high school students in Mexico. The internet-related experiences questionnaire (IREQ), the mobile-related experiences questionnaire (MREQ), and the Pittsburgh Sleep Quality Index (PSQI) were used. Results: The study revealed that 66% of participants exhibited some form of problematic internet use, primarily in the form of social media use; 68% had some form of problematic smartphone use, and 84% reported poor sleep quality. The PSQI score was most accurately predicted by problematic smartphone use (MREQ), followed by enrollment in the morning school shift, participation in sports, the father’s education level, and knowledge that “smartphone use disturbs sleep”, which together explained 23% of the variation in sleep quality. Conclusions: Excessive smartphone use may negatively affect sleep quality in adolescents. We recommended that interventions be implemented to educate adolescents about appropriate and healthy use of technology, in parallel with the promotion of preventive sleep habits.

## 1. Introduction

Adolescence is a time of growth, when many brain functions and personality traits are consolidated. Changes in brain regions involved in social processes can lead adolescents to focus intensely on peer relationships and social experiences [1]. During this stage, the value of social relationships and the pleasure they provide are very important in a teenager’s life, which is why electronic devices have become one of the most preferred means of socializing and have made young people avid users of social media [2].

In 2018, 70% of adolescents globally, between 13 and 17 years, accessed social media more than once a day, compared to 34% in the same study in 2012; furthermore, by 2018, 89% of teenagers had their own smartphone, compared to 41% of those who had a smartphone a decade earlier [3]. In Mexico, the estimated number of hours that users devote to social media ranges between eight and nine hours a day [4] and on average, they spend three hours a day in front of the computer or some other electronic device that allows them to connect to the internet. Likewise, it has been pointed out that 90% of young people say they cannot live without the internet, since it allows them to download music, videos and participate in social media [5]. In the United States, people aged 13 to 21, mostly adolescents (97%) were found to use some type of electronic device before going to sleep [6].

The consequences of excessive use of social media and smartphones are varied. Particularly, the use of electronic devices before sleeping has been associated with an increased risk of decreased sleep duration, deficiencies in the sleep pattern and prolonged latency at the onset of sleep [7]. Nighttime use of information and communication technologies (ICT, especially smartphones for which most users report problematic use) directly interferes with the circadian cycle, causing irregular sleep patterns [8,9]. The use of entertainment and communication technologies an hour before sleep is also negatively associated with sleep quality and efficiency. A 15 min deficiency in sleep duration can produce clinical effects; the greatest impact occurs when using a smartphone one hour before bedtime, which can decrease sleep quality by 36% [10]. Studies on sleep in adolescents show that, on average, for every hour that an adolescent goes to bed later, sleep onset latency is prolonged, and a shorter duration of sleep is generated (by approximately six and a half hours) on sleep days per week. The above brings about a deficit of approximately two hours of sleep per week [11,12,13,14]. Thus, increased technology use and frequency of being awakened by a smartphone during the night were significantly associated with waking up too early, not feeling refreshed upon waking, and having daytime sleepiness [7]. 

Sleep is essential to maintain the body’s general health due to its role in physiological and psychological recovery. Sleep is also associated with the consolidation of different forms of memory; it regulates temperature; fosters the function of certain neurotransmitters; stores energy; and maintains immunocompetence [15,16]. For these reasons, getting the highest quality sleep possible is crucial as it promotes good health. However, sleeping well not only requires complying with the recommended hours of sleep—which for adolescents includes between 8 and 10 h a day [17]—sleep must also be restorative, that is, it must be continuous and comply with the person’s circadian cycle and its different phases to reach depth [18]. However, among adolescents there is a prevalence of poor sleep quality of approximately 53%, which is associated with greater symptoms of depression and anxiety, as well as with lower academic performance [19,20].

Because sleep disturbances constitute a public health concern, and smartphones are now part of most adolescents’ lives, problematic use and inadequate sleep have been studied extensively. A recent meta-analysis involving 253,904 adolescents (mean [SD] age, 14.82 [0.83] years; 51.1% male) confirmed the unequivocal role of technology and the need to implement strategies to improve sleep quality and hygiene that include teachers, parents and health care providers [21]. Young people’s affinity for technology has also led researchers to search for moderators of the links between technology use and sleep to address why teenagers are sleeping too late or too little [22]. By contrast, excessive smartphone use is less likely to occur in students who participate in more physical activities and recreational sports [23].

Given the importance of sleep quality during adolescence, a stage particularly affected by biological factors [24,25] as well as changes in lifestyle dynamics, in which, to date, the use of smartphones is increasing, collecting data on this new area of research constitutes a public health priority. Especially, among public high school students in Mexico, where social and recreational activities are limited to the school environment and travel times to and from school are generally long and cumbersome, communication using smartphones is favored. Thus, the objective of this study was to explore if there was an association between problematic smartphone and internet use with poor sleep quality.

## 2. Materials and Methods

### 2.1. Participants and Procedure

Our study was an observational, cross-sectional and analytic design. Participating groups included students attending a public high school who were enrolled in their fifth semester taking the subject of Health Sciences II. In Mexico, most students attend public institutions, which are funded at the federal and state level. The population enrolled in these institutions generally comprises middle and low income families, some facing financial hardship. Despite this, in public schools, students have access to sports and cultural activities that allow them to have leisure spaces and a competitive academic curricula. Thus our study population was recruited in a public high school that aims to train students to pursue higher education. Due to the lack of public schools’ infrastructure, students in Mexico as in other countries in Latin America, use a double-shift school system attending a morning or an afternoon shift. One group of students comes early in the morning and leaves at mid-day (7 am to 1 pm), and a second group arrives at mid-day and leaves late in the evening (3 pm to 9 pm). A total of 190 high school students were enrolled in the study, 255 were invited to participate and our response rate was 75%. Their average age was 17.43 (±0.71) years, most were women (64%), and were full time students that did not work (78%). The proportion of students in the morning and afternoon shifts was similar (morning, 50%). Most parents of the study participants had completed college [mother (44%) and father (47%)]. We used non-probabilistic convenience sampling for all seven groups; three were enrolled in the morning and four in the afternoon shift.

Data collection was carried out using online surveys through the Google Forms platform. Parents were informed that an anonymous survey would be fielded and authorization was requested from parents and participants. They were also told the survey would include questions on use of smartphones and social media and its association with quality of sleep among students. A requirement for its completion was that the student had to select the option of informed consent of parents/guardians and their assent, to agree to take the survey as well as having to respond to all the questions before submitting it to avoid the loss of information or missing data. The survey was administered and personally supervised by one of the researchers between 13 and 20 February 2024 (mid-semester). Therefore, participants were asked to complete the survey during class time. On average it took them 20 min to complete all study instruments. 

### 2.2. Measures

#### 2.2.1. Collection of Personal Data

We asked about the students’ main sociodemographic characteristics (age, gender, and level of parental educational attainment) and an open-ended question about whether they participated in outdoor or recreational activities (e.g., reading, sports, socializing with friends, attending art classes, etc.). This question was coded as positive if participants indicated they engaged in any outdoor or recreational activity. We also created another variable if they mentioned that they practiced a sport, which was coded as yes and no.

#### 2.2.2. Problematic Internet Use

We administered the internet-related experiences questionnaire (IREQ), an instrument developed in Spain by Beranuy et al. [26]. The survey includes ten items that measures the type of interaction the user has when using the internet, answering every question using a Likert-type scale with four response options ranging from 1 = “not at all” to 4 = “quite a bit”. The IREQ includes questions such as: How often do you leave the things you are doing to spend more time connected to social media?

Its psychometric properties, evaluated in the study carried out by Beranuy et al., in 2009, show the questionnaire covers two factors: the first termed “intrapersonal conflicts” (items 4, 5, 6, 7, 9, 10) focused on intrapersonal consequences such as: focusing, worry, evasion, denial and other cognitive distortions, considered criteria to diagnose internet addiction. The second factor “interpersonal conflicts” (items 1, 2, 3, 8), evaluates the intrapersonal consequences of internet use which describes the progressive time invested and social–interpersonal conflicts that derive from the person’s online connections and addiction criteria that is associated with a decrease in other activities not related to internet use. The score is calculated by adding the responses to all the items, a high score reflects problematic internet use. Its Cronbach’s alpha reliability coefficient is 0.76 [27]. From the Spanish version, the authors reviewed the wording of the items to identify statements or words that were poorly understood in Mexican Spanish, and none were identified; the instrument was then piloted among 10 students who also reported no difficulties comprehending the wording of the questions. Thus, the Spanish version was used without the need to make any change or adaptation to the local context. In our study, the McDonald’s ω value was 0.74 for the total scale, and 0.70 and 0.58 for the intrapersonal conflicts and interpersonal conflicts scales, respectively.

#### 2.2.3. Problematic Smartphone Use

We administered the Smartphone-related experiences questionnaire (MREQ). It was also developed by Beranuy et al. [26] and includes ten items that assess the experiences associated with smartphone use. The survey is scored using a Likert-type scale that includes four options from 1 = “not at all”, to 4 = “quite a bit”. The MREQ includes questions such as: Do you think that your academic or work performance has been negatively affected by your smartphone use? The score is calculated by adding the responses to all the items, until we obtain a maximum score of 40; a high score reflects problematic smartphone use. The questionnaire covers two factors: “conflict” and “communicational and emotional use”. The conflict factor was defined as the conflicts that arise intrapersonally, such as inability to focus, worry, evasion, denial; and interpersonally (progressive increase in time devoted to smartphone use and social–interpersonal conflicts) regarding the use of the smartphone (items 1, 2, 4, 5, 8). Communicational and emotional use was defined as the evaluation of problematic communication, since smartphone abuse seems to be characterized by an impulsive pattern of use (items 3, 6, 7, 9, 10). The MREQ internal consistency using Cronbach’s alpha coefficient is 0.79 [27]. The Spanish version was reviewed and subsequently piloted on 10 students to evaluate its comprehension; participants showed no difficulties understanding the questions. In our study, the McDonald’s ω value was 0.79 for the total scale, with ω value of 0.70 and 0.58 for the conflict and communicational and emotional use scales, respectively. 

In addition, with the aim of exploring if participants were aware of the impact of their smartphone use at night, we included an additional question, “Do you think that using your smartphone before bedtime interferes with your quality of sleep?”.

#### 2.2.4. Sleep Quality

The Pittsburgh Sleep Quality Index (PSQI) is a self-assessment instrument originally developed by Bussye, in 1989 [28]; it is widely used in clinical practice and research in Mexico and internationally. The PSQI measures sleep quality over a period of one month; it includes 19 items grouped into seven components: (1) sleep quality, (2) sleep latency, (3) sleep duration, (4) habitual sleep efficiency, (5) sleep disturbance, (6) use of sleeping medications and (7) daytime dysfunction. The PSQI items record the participant’s usual bedtime, usual waking time, number of actual hours slept, and the number of minutes it takes for the person to fall asleep as well as the subject’s appreciation of some sleep problems. Responses are coded using a Likert scale that measures frequency or severity [29]. The instrument has an internal consistency for all 19 items of 0.83 (Cronbach’s alpha) [28]. Research on sleep disturbance shows the PSQI compares favorably with results obtained in polysomnography; when we obtain a score of >5, the instrument displays a sensitivity of 89.6% and a specificity of 86.5%, to indicate significant sleep disturbance [30]. We used the Spanish version of this scale adapted to Mexico whose reliability among Mexican university students, evaluated by Cronbach’s alpha was 0.79, and a convergent validity with the Athens Insomnia Scale of 0.71 [31]. In our study sample, the McDonald’s ω value was 0.79.

### 2.3. Data Analysis

Statistical analysis was conducted using SPSS Statistics (IBM, version 29). Descriptive statistics were employed to characterize sleep quality, problematic internet use (IREQ), and problematic smartphone use (MREQ) in our study population. Frequencies and percentages were utilized to describe the levels observed, whereas means and standard deviations were employed to report scale scores. Bivariate correlations using the Spearman test were performed between the PSQI (total and components) and participants’ sociodemographic characteristics, as well as with the total scores of the IREQ and the MREQ. The components and total PSQI were compared using the Mann–Whitney U test when variables were categorical. The main factors associated with sleep quality were also assessed using a robust generalized linear model, with sociodemographic characteristics, problematic internet use (IREQ) and problematic smartphone use (MREQ) as independent variables. This type of analysis is a more recommended option than linear regression when there are exogenous categorical variables [32]. Hypothesis tests used were two-tailed and significance was set at *p* < 0.05.

## 3. Results

Table 1 describes the characteristics of sleep, internet, and smartphone use. Of the participants, 84% reported poor sleep quality, 66% had occasional or frequent problems with internet use, and 68% had occasional or frequent problems with smartphone use. The average number of hours per day spent on social media was 4.6 (±3.6) hours, while the average number of hours per day spent on a smartphone was 6.4 (±3.6). The most common use for their smartphone was social media (including visiting entertainment sites) among 74% of participants. The average age at which students first began to use a smartphone was 11.05 ± 2.3 years.

Regarding the subjective quality component, significant differences were observed when students engaged in recreational (reading, sports, socializing with friends, attending art classes, etc.), participated in outdoor activities, and knew that using smartphones at night disturbs sleep (Table 2). We also found a significant correlation with the father’s level of education, problematic internet use, and problematic smartphone use (Table 3). Sleep latency showed differences among students that knew using a smartphone at night disturbs sleep and those who did not (Table 2) and was related to the time using their smartphone and time spent on social media, problematic internet use, and problematic smartphone use (Table 3). The duration was influenced by the participant’s school shift and sex (Table 2) and showed a relationship with the time spent on social media, problematic internet use and problematic smartphone use (Table 3). Efficiency did not show an association with any of these variables.

Sleep disturbances were associated with the mother’s level of educational attainment, problematic internet use, and problematic smartphone use (Table 3). Medication use (sleep aids) only correlated with the father’s level of education (Table 3). Daytime dysfunction was influenced by engaging in outdoor or recreational activities and knowing that using a smartphone at night disturbs sleep (Table 2), and was related to problematic internet use, and problematic smartphone use (Table 3). Finally, the student’s school shift, engaging in recreational or outdoor activities, agreeing that smartphone use at night disturbs sleep showed an effect on the total score of sleep quality (Table 2) were related to the time spent on social media, smartphone use time, problematic internet use and problematic smartphone use (Table 3).

Multivariate analysis (generalized linear model) showed that the main factors associated with an increase in the PSQI (worse sleep quality) were that the student was enrolled in the morning shift, had a higher score on the MREQ, and knew that “smartphone use alters sleep”; while a higher academic attainment in the father and practicing a sport were associated with a decrease in the PSQI score (χ^2^(6)= 54.43, *p* < 0.001). See Table 4.

## 4. Discussion

The aim of our study was to examine whether there is an association between problematic smartphone and problematic internet use with poor sleep quality. This finding was partially confirmed, as only problematic smartphone use together with other participant characteristics showed an association with sleep quality. We found a high prevalence of poor sleep quality among students, as well as high levels of problematic smartphone and internet use. This study sets a precedent regarding the prevalence of poor quality of sleep, problematic use internet and smartphone use in high school students in Mexico City. Similar studies have been fielded mainly in university students. Our findings could open up new research in other public high schools that are geographically different and with a different educational context from those in the present study. 

More than eighty percent of students in our study were classified as poor sleepers, with an average PSQI score of over six points and an average of six hours slept. A similar finding was reported among Scottish secondary school students (aged 11–17 years) who reported an average of 5.3 (±3.2) on the PSQI [33]; this value, despite exceeding the PSQI cut-off point for sleep quality, is not as high as the one found in our study. On the other hand, another study from Brazil, using the PSQI among adolescents found that 53% of participants reported having poor sleep quality and adolescents who were at high risk of depression were 3.45 times more likely to also report poor quality of sleep [20]. These findings reflect the current concern and need to improve the duration and quality of sleep in adolescent students. 

The factors that best explained the participants’ sleep quality included higher scores in problematic smartphone use, being enrolled in the morning shift at school and knowing that smartphone use at night disturbs sleep, which worsened their sleep quality, while their sleep quality improved when they reported practicing sports and having a father with higher educational attainment. Regarding the first point, meta-analytic studies have linked addictive smartphone use to sleep quality problems as well as emotional problems in university students [34]. Specifically, among adolescents in Scotland, researchers found that using social media (on any electronic device) at night was associated with poorer sleep quality [33]. In this regard, the increased use of technology and frequency of being woken up by a smartphone during the night were significantly associated with waking up too early, not feeling refreshed upon waking, and daytime sleepiness; adolescents who reported having “inadequate” sleep had a shorter duration of sleep period, greater frequency of technology use before bed, did not feel refreshed upon waking, and had greater daytime sleepiness compared with those who reported having “adequate” sleep” [6]. Adolescents with problematic smartphone use, including the habit of spending ample time in front of screens, can also show negative effects on their sleep cycle. This may be because light, especially short-wavelength light (blue light), emitted by screens can alter circadian processes such as the release of melatonin [9,10]. We believe that internet use was not considered significant in our multiple analysis because both aspects may be related, and adolescents may be more aware of smartphone use than internet use.

Other factors associated with poorer sleep quality in our study include the school shift in which students were enrolled, on the one hand; and knowing that smartphone use at night disturbs sleep, on the other. Regarding the school shift, we found that students who attended school in the morning had overall significantly poorer sleep quality and a worse sleep duration score. This can be explained by the fact that adolescents show a delay in their sleep phase [24] and, in the case of our morning shift students, because classes begin at 7 a.m., this significantly reduces the number of hours they are able to sleep. Educational experts have raised the need for a later school start time (between 8:30 and 9:45 a.m.), as this favors the natural sleep schedule of adolescents and allows them to sleep longer [35,36]. It has also been reported that adolescents who start school later report adequate sleep compared to adolescents who attend a fixed morning school schedule [37]. However, these same authors point out that this can also be modified by adolescent chronotype. Adolescents who start school later report adequate sleep compared to adolescents who attend a fixed morning school schedule. On the other hand, it is noteworthy that participants in our study were aware and knew that “exposure to smartphones at night alters sleep” and that this worsened their sleep quality. Knowledge of sleep habits per se does not improve its quality, but rather its practice [38], and adolescents tend to have a more negative attitude toward the importance of sleep, which affects their sleep hygiene practices [39]; we believe that the mechanism by which this association takes place merits further research.

Among the factors that promote sleep quality, we found an association between practicing sports and the father’s educational attainment. First, the benefits of physical activity on sleep have been widely recognized, including among adolescents [40,41,42,43,44]. In this regard, participating in sports has been associated with better sleep; for example, adolescents who practice sports more than three times a week are more likely to meet the recommended sleep duration [45] and display fewer sleep disorders, including bruxism that has been reported in children who play sports [46]; in college, team sports have been shown to have a positive effect on sleep quality and depression [47]. There may also be an indirect way sleep plays a role, since getting involved in sports has also been associated with decreased smartphone use which decreases the interference of devices on sleep [23], while the role of the father’s educational attainment allows us to draw conclusions about his role in the sleep of his offspring. On the one hand, it has been suggested that perceived parental support (i.e., having bedtime rules) is strongly associated with sleep duration and/or sleep quality in adolescents [48]. However, only one study found an association between the parent’s educational level with adolescents’ sleep. This study reported that parental control, construed as a negative interaction style, seems to have an impact on excessive use of smartphones, noting that higher parental educational attainment was associated with less parental control, greater satisfaction of children’s needs, less frustration of needs and less problematic smartphone use [49]. However, as some adolescents may still be under parental control, some studies made on children shed some light on the possible mechanism that educational attainment plays on the quality of sleep.

In children, parental education has been shown to influence the presence of sleep problems in more disadvantaged populations such as Latinos and African Americans, but not in Caucasians, perhaps suggesting that a lower educational level is a form of social disadvantage associated with less parental control [50]. 

We also observed that more than half of our students reported problematic smartphone and problematic internet use, which in most of them was spent on social media. This problematic use or abuse of smartphones (with an average of six hours per day among participants) added to the poor quality of their reported sleep which may also directly affect their academic performance. Technology and the digital world where adolescents now socialize are part of the tools we use in our daily lives. However, problematic use can quite easily become addictive behavior, as one depends on the other, given that smartphones have increased and eased our internet access, and the frequency and popularity of social media have made it the new normality for adolescent interaction. Over the years, we have witnessed a trend where sleep problems among adolescents, and particularly students, is increasing. A study carried out in Spain among adolescents between 12 and 17 years, described the psychosocial profile of adolescents with problematic use of new technologies. The authors describe that maladaptive use can be found among people who show life dissatisfaction, poor family support, introversion, negative thoughts, discomfort with real social interactions and conflict with identity. The researchers found that smartphones seem to be the technology with the most maladaptive use compared to internet use, especially when it is used as a communication and emotional mode of communication [51]. Young people who do not enjoy effective communication with their parents tend to spend more hours online, thus making up for the absence of communication at home. This problem was also highlighted by younger participants (15 and 17 years old), who acknowledged the discomfort caused by not being able to connect to the internet for several days, and this was more pronounced when being offline, preventing them from accessing social media [52]. 

Smartphones have increased the ease of connectivity to the internet and the frequency with which social media can be accessed has popularized this new way of socializing for young people. Regarding the frequency of social media use, we found that those most used by adolescents in this study were, in descending order, TikTok, Instagram and WhatsApp. The way in which short videos from TikTok and Instagram are viewed causes the user to go from one subject to another very quickly causing this consumption to last longer than intended. The WhatsApp also plays a role in this vicious cycle, by keeping users pending messages or notifications at night. Due to its design, which is deliberately made to be addictive because its algorithm uses machine learning to know what content everyone prefers based on their behavior [53], adolescents may prefer it because the content presented by these applications is of short duration, not requiring much concentration on a specific topic and producing hooking.

## 5. Conclusions

The present study is limited by our use of subjective measures. It should be noted that participants tend to underestimate their responses in self-assessment surveys, which may result in lower recorded results. Additionally, in the case of sleep, the assessment did not consider whether students were taking exams at the time of the survey, which may disrupt their normal sleep cycle due to overwork. The causal relationship between sleep quality, problematic internet use, and problematic smartphone use cannot be established by this study due to its cross-sectional nature. Some of our study variables were not analyzed in depth such as number of hours participants practiced sports or spent in leisure activities. However, this is an important area that should be investigated more in depth. Additionally, due to the type of selection of our sample, the generalizability of our results is also limited.

Based on these considerations, we can thus conclude that according to our study aims, the time that students spend using their smartphone or accessing the internet was high and this was the strongest factor associated with sleep quality, controlling for the school shift they were enrolled in, their father’s educational attainment, playing sports, and knowing that smartphone use at night affects sleep. Our findings point to the urgent need to find ways of reducing dependence on smartphones and social media. Some recommendations that could help mitigate the above include establishing time limits for smartphone use (digital hygiene), helping to raise awareness of the time that young people spend online, and establishing digital media-free moments, for example, during meals or before sleeping. Likewise, alternative activities should be promoted so young people can stay away from smartphones, such as practicing sports, reading or spending time with friends and family, which can help reduce anxiety and improve the quality of free time and thus have positive health benefits. As there are no subjects in the current academic curricula of the main public high schools in Mexico that address these prevention issues, it is of utmost importance to act at the high school level or perhaps earlier. Interventions to educate adolescents about appropriate and healthy use of technology, in parallel with promoting preventive sleep habits, can mitigate this problem that increasingly afflicts adolescents worldwide. Surgeon General Vivek Murthy writes in Opinion: “It is time to require a surgeon general’s warning label on social media platforms, stating that social media is associated with significant mental health harms for adolescents”, from the Times, 17 June 2024, a statement that stands in agreement with our study findings.

## Figures and Tables

**Table 1 ijerph-21-01177-t001:** Characteristics of smartphone use, internet use, and sleep among participants.

Sleep	
Sleep quality level ^a^	
Poor	159 (84)
Good	31 (16)
Sleep quality (score) ^b^	7.5 ± 3.2
Sleep duration (hours) ^b^	6.1 ± 1.5
Internet and smartphone use	
Age first use of smartphone (years) ^b^	11.0 ± 2.3
Time using smartphone (hours per day) ^b^	6.4 ± 3.9
Knowledge that “smartphone use at night disturbs sleep” ^a^	
Yes	135 (71)
No	55 (29)
Time spent on social media (hours per day) ^b^	4.6 ± 3.6
Social media platform ^a^	
TikTok	69 (36)
Instagram	57 (30)
WhatsApp	39 (20)
YouTube	10 (5.5)
Facebook	10 (5.5)
Other (X or Pinterest)	5 (3)
Main internet access ^a^	
Home internet	178 (94)
Cellular network	10 (5)
Free WIFI	2 (1)
Problematic internet use (level) ^a^	
No problem	65 (34)
Occasional problems	107 (56)
Frequent problems	18 (10)
Problematic internet use (IREQ score) ^b^	17.8 ± 4.4
Problematic mobile-phone use (level) ^a^	
No problem	60 (32)
Occasional problems	109 (57)
Frequent problems	21 (11)
Problematic smartphone use (MREQ score) ^b^	19.4 ± 4.4
Most common use of smartphone ^a^	
School work	43 (22)
Calls	7 (4)
Social media	140 (74)

^a^ Data described with frequencies (percentages). ^b^ Data described with means ± standard deviation.

**Table 2 ijerph-21-01177-t002:** Comparisons of subjective sleep quality and its components between the main study variables.

	PSQI Components														Total	*Z*
	Quality	*Z*	Latency	*Z*	Duration	*Z*	Efficiency	*Z*	Disturbance	*Z*	Sleep Medication	*Z*	Daytime Dysfunction	*Z*		
Sex		−0.28		−0.16		−2.29 *		−0.17		−0.09		−0.33		−1.67		−0.99
Male	1.2 ± 0.7		1.2 ± 0.9		1.3 ± 1.0		0.6 ± 1.0		1.1 ± 0.5		0.2 ± 0.7		1.4 ± 0.8		7.1 ± 3.1	
Female	1.2 ± 0.8		1.3 ± 0.9		1.6 ± 0.9		0.6 ± 1.1		1.1 ± 0.5		0.3 ± 0.6		1.7 ± 0.9		7.7 ± 3.3	
School shift		−0.31		−0.53		−6.69 ***		−1.06		−0.76		−0.38		−1.60		−2.16 *
Evening	1.2 ± 0.8		1.3 ± 0.9		1.0 ± 0.9		0.7 ± 1.0		1.1 ± 0.4		0.2 ± 0.6		1.5 ± 0.8		7.0 ± 3.3	
Morning	1.2 ± 0.7		1.2 ± 0.9		2.0 ± 0.8		0.5 ± 1.0		1.2 ± 0.6		0.3 ± 0.7		1.7 ± 0.9		7.9 ± 3.2	
Participates in outdoor or recreational activities		−3.29 ***		−1.61		−0.64		−1.07		−0.91		−0.63		−2.28 *		−2.65 **
No	2.1 ± 0.8		1.8 ± 1.0		1.7 ± 1.0		1.1 ± 1.0		1.0 ± 0.0		0.3 ± 0.7		2.2 ± 0.7		10.2 ± 2.5	
Yes	1.4 ± 0.7		1.2 ± 0.9		1.4 ± 1.0		0.6 ± 1.0		1.1 ± 0.4		0.2 ± 0.6		1.6 ± 0.9		7.3 ± 3.2	
Practices sports		−1.50		−0.02		−1.10		−0.76		−1.03		−0.79		−1.64		−1.42
No	1.3 ± 0.8		1.3 ± 1.0		1.6 ± 1.0		0.7 ± 1.1		1.2 ± 0.4		0.3 ± 0.8		1.7 ± 0.9		8.0 ± 3.5	
Yes	1.1 ± 0.7		1.2 ± 0.9		1.4 ± 0.9		0.6 ± 0.9		1.1 ± 0.5		0.2 ± 0.5		1.5 ± 0.9		7.1 ± 3.0	
Knowledge that “smartphone use at night disturbs sleep”		−2.21 *		−2.18 *		−0.90		−0.03		−1.46		−0.43		−2.59 **		−2.01 **
No	1.0 ± 0.6		1.0 ± 0.8		1.4 ± 0.9		0.6 ± 0.9		1.1 ± 0.4		0.2 ± 0.7		1.3 ± 0.8		6.6 ± 2.6	
Yes	1.3 ± 0.8		1.3 ± 0.9		1.5 ± 1.0		0.6 ± 1.0		1.2 ± 0.5		0.2 ± 0.6		1.7 ± 0.9		7.8 ± 3.5	

Note: A higher score on the PSQI or its components indicates a poorer response to sleep. *** *p* < 0.001, ** *p* < 0.01 * *p* < 0.05.

**Table 3 ijerph-21-01177-t003:** Bivariate correlations of the main study variables with subjective sleep quality and its components.

	PSQI Components							Total
	Quality	Latency	Duration	Efficiency	Disturbance	Sleep Medication	Daytime Dysfunction	
Age (years)	−0.13	0.00	−0.11	0.04	−0.04	−0.10	−0.08	−0.09
Age first use of smartphone (years)	−0.03	−0.12	−0.11	−0.04	−0.10	−0.01	−0.03	−0.10
Mother’s educational attainment	−0.07	0.01	0.05	−0.07	−0.16 *	−0.13	−0.08	−0.09
Father’s educational attainment	−0.16 *	−0.09	0.05	0.02	−0.05	−0.15 *	−0.07	−0.10
Time using smartphone (hours per day)	0.12	0.22 **	0.13	0.11	0.11	−0.05	0.07	0.18 *
Time spent on social media (hours per day)	0.12	0.18 *	0.17 *	0.07	0.12	−0.00	0.13	0.20 **
Problematic internet use (IREQ score)	0.25 ***	0.25 ***	0.17 *	−0.00	0.25 ***	0.01	0.18 *	0.28 ***
Problematic smartphone use (MREQ score)	0.33 ***	0.33 ***	0.20 **	0.09	0.23 **	0.05	0.34 ***	0.40 ***

Note: A higher score on the PSQI or its components indicates a poorer response to sleep. *** *p* < 0.001, ** *p* < 0.01 * *p* < 0.05.

**Table 4 ijerph-21-01177-t004:** Factors associated with sleep quality (PSQI).

	B	Std. Error	95% Wald CI	Wald χ^2^	*p*
Intercept	3.04	0.92	1.24, 4.85	10.91	<0.001
School shift (morning)	1.00	0.40	0.21, 1.80	6.13	0.013
Practicing a sport (yes)	–0.82	0.40	–1.61, –0.03	4.15	0.042
Knowledge that “smartphone use at night disturbs sleep” (yes)	0.86	0.40	0.08, 1.64	4.68	0.030
Father’s educational attainment	−0.46	0.20	–0.85, –0.07	5.26	0.022
Problematic smartphone use (MREQ score)	0.29	0.04	0.20, 0.38	43.35	<0.001

Note. A higher score on the PSQI or its components indicates a poorer response to sleep. CI: confidence interval.

## Data Availability

The datasets used and/or analyzed during the current study are available from the corresponding author on reasonable request.

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
