# Peer review of "Problematic Use of Smartphones and Social Media on Sleep Quality of High School Students in Mexico City"

_ijerph, 2024, doi:10.3390/ijerph21091177_

Round 1
Reviewer 1 Report
Comments and Suggestions for Authors
The manuscript presents the results of an interesting study aiming to find out the problematic use of mobile phones and social media on sleep quality of high school students. Some considerations that could help to improve it:
1. The theoretical framework is adequate, and is based on relevant and current bibliographical references. However, the information corresponding to the relationship between the two main study variables: use of mobile phones (types of applications, modulating variables...) and sleep quality could be expanded. In fact, there is more information on this in the discussion, which is not referred to in this section. At one point, reference is made to the significance of a result (p < .05) on p.2, line 66. It is already said to be significant, so it will necessarily be below that level of significance. It would be different if reference were made to effect sizes (where the amount of the value itself gives information on the amount of difference), but significance only informs whether or not it is significant.
2. Participants: Participants belong to one school. Since they all come from the same context, it would be convenient to describe this context, given that the results may be conditioned by the socio-economic characteristics of the school population. On the other hand, this is a limitation that should be referred to in the corresponding section.
3. Measures: Information on the internal consistency of the scales used is provided using the score of the sample under study. For this reason, it would be convenient to use the McDonald omega statistic, given that it is currently recognised that it is a better fit with ordinal variables (such as Likert scale items) than Cronbach's Alpha, which is more suitable for continuous quantitative variables, and which tends to overestimate its value when applied to ordinal variables and with large samples. On the other hand, the internal consistency values of the scale totals are provided. However, all of them have factors. Internal consistency should be provided for each of the factors of each scale.
4. Data analysis: The calculation of Spearman's correlation coefficient is appropriate when the assumption of normality is not met, or the variables to be compared are not continuous, but ordinal, but not when they are categorical (e.g. gender). The same applies to the independent variables in the multiple regression analysis (it could be done with dependent variables by making a dummy choice, although this is not the case).
5. Results: Table 1 and the corresponding text should not go in the results section. The information corresponds to the description of the participants, so it should be moved to section 2.1. As mentioned above, not all correlations can be calculated. This is acceptable, though doubtful with some of the ordinal variables (such as educational level, given the ‘conceptual distance’ that may exist between categories), but not with categorical variables that do not involve ordering (e.g. sex). In these cases, why not use tests of mean or rank differences, either parametric or non-parametric respectively, such as Student's t-test or Mann-Whitney U test, or ANOVA...?
6. Limitations are not usually included in the discussion section. They are usually in the conclusions.
Author Response
|
Comment |
Response |
Location of the change |
|
1. The theoretical framework is adequate and is based on relevant and current bibliographical references. However, the information corresponding to the relationship between the two main study variables: use of mobile phones (types of applications, modulating variables...) and sleep quality could be expanded. In fact, there is more information on this in the discussion, which is not referred to in this section. At one point, reference is made to the significance of a result (p < .05) on p.2, line 66. It is already said to be significant, so it will necessarily be below that level of significance. It would be different if reference were made to effect sizes (where the amount of the value itself gives information on the amount of difference), but significance only informs whether or not it is significant. |
Thank you for your observations.
We have thus rewritten the text to clarify this further in the introduction.
We also agree with your other suggestion and have deleted the (p < .05) only mentioning results were relevant and a reference. |
Lines
76-85
Line 64
|
|
2. Participants: Participants belong to one school. Since they all come from the same context, it would be convenient to describe this context, given that the results may be conditioned by the socio-economic characteristics of the school population. On the other hand, this is a limitation that should be referred to in the corresponding section. |
The educational context of the study population was added.
|
Lines 99-105 |
|
3. Measures: Information on the internal consistency of the scales used is provided using the score of the sample under study. For this reason, it would be convenient to use the McDonald omega statistic, given that it is currently recognized that it is a better fit with ordinal variables (such as Likert scale items) than Cronbach's Alpha, which is more suitable for continuous quantitative variables, and which tends to overestimate its value when applied to ordinal variables and with large samples. On the other hand, the internal consistency values of the scale totals are provided. However, all of them have factors. Internal consistency should be provided for each of the factors of each scale. |
As suggested, McDonald omega statistics were calculated for the total scales and the instrument subscales.
|
Lines 156-158, 176-178, 198-199 |
|
4. Data analysis: The calculation of Spearman's correlation coefficient is appropriate when the assumption of normality is not met, or the variables to be compared are not continuous, but ordinal, but not when they are categorical (e.g. gender). The same applies to the independent variables in the multiple regression analysis (it could be done with dependent variables by making a dummy choice, although this is not the case). |
For categorical variables, the analysis using correlations was eliminated and the use of the Mann-Whitney U test was changed.
Linear regression was changed to an analysis using a generalized linear model, in which categorical variables are analyzed as groups
|
Lines 208-209 Table 2
Lines 209-213 Table 4 |
|
5. Results: Table 1 and the corresponding text should not go in the results section. The information corresponds to the description of the participants, so it should be moved to section 2.1. As mentioned above, not all correlations can be calculated. This is acceptable, though doubtful with some of the ordinal variables (such as educational level, given the ‘conceptual distance’ that may exist between categories), but not with categorical variables that do not involve ordering (e.g. sex). In these cases, why not use tests of mean or rank differences, either parametric or non-parametric respectively, such as Student's t-test or Mann-Whitney U test, or ANOVA...? |
The information from the participants was moved to the method.
The correlation analysis of the categorical variables was replaced by comparison tests (Mann-Whitney U test).
|
Lines 109-114
Table 2 |
|
6. Limitations are not usually included in the discussion section. They are usually in the conclusions. |
The limitations of the study were moved to the conclusions section.
|
Lines 392-402 |

Reviewer 2 Report
Comments and Suggestions for Authors
The following comments regard the manuscript entitled “Problematic Use of Mobile Phones and Social Media on Sleep Quality of High School Students in Mexico City”. Authors wrote the manuscript paying great attention to scientific aspects; this deserves to be emphasized. However, based on my knowledge of the topic, this work presents two points of improvement on theoretical and methodological levels.
POINT OF IMPROVEMENT: I think that this work should be improved in terms of positioning in the current scenarios of scientific literature, to suggest more specific comprehension of the way to reduce Internet and technology use in the time before the sleep.
AUTHORS POSITION: Authors report that “The study sets a precedent regarding the prevalence of poor quality of sleep, problematic use Internet and mobile phone use in high school students in Mexico. Similar studies have been fielded mainly in university students. Our findings could open new research in other public high schools geographically different and with a different educational context from those of the present study”.
REVIEWER’S COMMENT: Some current literature recognizes technology use as a precursor to sleep quality (and quality of life in general) in samples of adolescents. In order to gain a better understanding of the topic to implement strategies to contrast the maladaptive use of the Internet and technology, could the authors clarify which aspects of this study integrate the current knowledges on the topic? If authors recognize an advancement of current knowledge in the selected sample (for example, students from Mexico); I believe they should better specify what aspects distinguish these sample such that the results would be specific and therefore useful to verify in other populations.
POINT OF IMPROVEMENT: Linked to the first point, the study presents a methodological framework introducing variables that - if configured differently - bring novelty to the current literature.
AUTHORS POSITION: Authors report “We ask the main sociodemographic characteristics of students (age, sex and parental education), and an open question aimed at investigating their participating in outdoor or recreational activities (e.g., reading, sports, socializing with friends, art, etc.)”; Problematic Internet and Mobile use were detected by the Internet-related experiences questionnaire (IREQ) and the mobile phone-related experiences questionnaire (MREQ); Sleep quality by The Pittsburg Sleep Quality Index (PSQI).
REVIEWER’S COMMENT: Authors use measures consistent with the purpose of the article. However, a point of innovation is found in the presence/absence of recreational activities. I believe that this aspect should be enhanced; even more so for what is reported in the previous point of improvement (i.e., specific sample). Investigating the presence/absence of recreational activities leads to reflecting on the importance of the context (and social cotext) within which technology creeps in. Although today we are all connected, some become problematic users, but not others. Why? In this sense, could the resources of the context offer a prevention strategy for the use of technology? I hypothesize two scenarios: a context without social resources; a second scenario rich in social resources. In the first, technology would be a mediator for socialization and emotional states (es. Loneliness); in the second case, the commitment to recreational activities would limit the use of the smartphone (or the Internet in general). On a methodological level, this leads to a different choice. Since all teenagers are connected, could the variable related to recreational and/or outdoor activities play a mediating/moderating role in the relationship between Internet use and sleep quality? From a theoretical point of view, this would lead to the need to further specify the context to which the recruited sample belongs (e.g. how is it geographically located? what are the habits? are there for example theatres/cinemas…? are there recreational spaces? what are the mainly available recreational activities?).
***
Further major points that deserve attention:
The results are well connected to the literature, but they seem like "pieces of a puzzle" that I struggle to put together to have an overall view of the study.
1) Authors report “It can be observed that 84% of participants reported poor sleep quality”. In the table 2, it can be seen that 84% is referred to “good” sleep quality. Am I seeing wrong? This information is reported also at row 231. Please, check it.
2) Authors report “Other factors associated with poorer sleep quality include the school shift in which students were enrolled and the knowledge that mobile phone use at night disturbs sleep. Regarding the school shift, educational experts have raised the need for a later school start time (between 8:30 and 9:45 a.m.), as this favors the natural sleep schedule of adolescents and allows them to sleep longer [32,33]. It has also been reported that adolescents who start school at a later time report adequate sleep compared to adolescents who attend a fixed morning school schedule [34]”. Could the authors better explain their position through the results of their study? In my opinion, it is not very clear how the authors use this information to argue that technology use before bed impacts the quality of the daytime hours.
3) Author report “The role of the father’s educational level allows us to draw conclusions about his role in the sleep of his offspring. On the one hand, their involvement may be indirect, through excessive use of mobile devices, as it has been reported that an insecure parental attachment style leads to problematic use [45], and parental control also seems to have an impact on excessive use of these devices [46]”. Subsequently, authors add this argumentation “A study that looked at the characteristics of both parents for insufficient sleep during childhood found that the risk of poor sleep was higher when mothers (but not fathers) worked 35 hours or more per week, and the risk of children not getting enough sleep was higher when the father worked in construction and production [47]”. Could the authors explain how the two statements are connected? While the second statement seems understandable to me, I find it difficult to understand the relationship between the parent's attachment style, their level of education, and the quality of their children's sleep.
Minor comments:
1) Please, put a double colon after "method" (row 14).
2) In the keywords, I suggest reporting problematic Internet use first and then problematic mobile phone use (row 28).
3) Why do the authors report the p-value on line 63 rather than the reference of the statement?
4) Did the authors use versions translated of instruments into the native language of Mexico? If so, could you please specify; if not, could you please describe the translation procedure. Then, it could be useful for reader to report some examples of items.
5) Descriptive characteristics of the sample could be reported in the sample paragraph rather than in the results.
I am very happy to have read this work; I hope that authors can take the aspects suggested in the comments as ideas to improve the applicability of these results. I apologize for the level of criticality, and I hope that – valorized the potential components of the work – this article can be successful.
Comments on the Quality of English LanguageI advise the authors to do a general check of the English even if overall it seems well written.
Author Response
|
Comment |
Response |
Location of the change |
|
POINT OF IMPROVEMENT: I think that this work should be improved in terms of positioning in the current scenarios of scientific literature, to suggest more specific comprehension of the way to reduce Internet and technology use in the time before the sleep. AUTHORS POSITION: Authors report that “The study sets a precedent regarding the prevalence of poor quality of sleep, problematic use Internet and mobile phone use in high school students in Mexico. Similar studies have been fielded mainly in university students. Our findings could open new research in other public high schools geographically different and with a different educational context from those of the present study”. REVIEWER’S COMMENT: Some current literature recognizes technology use as a precursor to sleep quality (and quality of life in general) in samples of adolescents. In order to gain a better understanding of the topic to implement strategies to contrast the maladaptive use of the Internet and technology, Could the authors clarify which aspects of this study integrate the current knowledges on the topic? If authors recognize an advancement of current knowledge in the selected sample (for example, students from Mexico); I believe they should better specify what aspects distinguish these sample such that the results would be specific and therefore useful to verify in other populations. |
Thank you for your observations. We added a bit of school context in Mexico, both in the introduction and in the method section.
|
Lines 89-92
Lines 99-105 |
|
POINT OF IMPROVEMENT: Linked to the first point, the study presents a methodological framework introducing variables that - if configured differently - bring novelty to the current literature. AUTHORS POSITION: Authors report “We ask the main sociodemographic characteristics of students (age, sex and parental education), and an open question aimed at investigating their participating in outdoor or recreational activities (e.g., reading, sports, socializing with friends, art, etc.)”; Problematic Internet and Mobile use were detected by the Internet-related experiences questionnaire (IREQ) and the mobile phone-related experiences questionnaire (MREQ); Sleep quality by The Pittsburg Sleep Quality Index (PSQI). REVIEWER’S COMMENT: Authors use measures consistent with the purpose of the article. However, a point of innovation is found in the presence/absence of recreational activities. I believe that this aspect should be enhanced; even more so for what is reported in the previous point of improvement (i.e., specific sample). Investigating the presence/absence of recreational activities leads to reflecting on the importance of the context (and social context) within which technology creeps in. Although today we are all connected, some become problematic users, but not others. Why? In this sense, could the resources of the context offer a prevention strategy for the use of technology? I hypothesize two scenarios: a context without social resources; a second scenario rich in social resources. In the first, technology would be a mediator for socialization and emotional states (es. Loneliness); in the second case, the commitment to recreational activities would limit the use of the smartphone (or the Internet in general). On a methodological level, this leads to a different choice. Since all teenagers are connected, could the variable related to recreational and/or outdoor activities play a mediating/moderating role in the relationship between Internet use and sleep quality? From a theoretical point of view, this would lead to the need to further specify the context to which the recruited sample belongs (e.g. how is it geographically located? what are the habits? are there for example theatres/cinemas…? are there recreational spaces? what are the mainly available recreational activities?). |
Some lines were included about the effect of recreational activities on cell phone use.
The context of access to recreational activities is indicated
|
Lineas 83-85
Lineas 99-105 |
|
*** Further major points that deserve attention. |
|
|
|
1) Authors report “It can be observed that 84% of participants reported poor sleep quality”. In the table 2, it can be seen that 84% is referred to “good” sleep quality. Am I seeing wrong? This information is reported also at row 231. Please, check it. |
The order of the information was corrected, it was reversed.
|
Table 2 |
|
2) Authors report “Other factors associated with poorer sleep quality include the school shift in which students were enrolled and the knowledge that mobile phone use at night disturbs sleep. Regarding the school shift, educational experts have raised the need for a later school start time (between 8:30 and 9:45 a.m.), as this favors the natural sleep schedule of adolescents and allows them to sleep longer [32,33]. It has also been reported that adolescents who start school at a later time report adequate sleep compared to adolescents who attend a fixed morning school schedule [34]”. Could the authors better explain their position through the results of their study? In my opinion, it is not very clear how the authors use this information to argue that technology use before bed impacts the quality of the daytime hours. |
We assume that being on the morning shift decreases the quality of sleep. However, we do not have elements to argue that the use of technology before sleeping alters the quality of sleep, since this particular aspect was not investigated. We only asked if they knew that using the cell phone before going to sleep altered sleep (which showed results contrary to what was hypothesized). To separate these two points, we revised the wording of the section in order to correct the confusion.
|
Lineas 310-316
Tables 1, 2 and 4 Lines 23, 231, 253, 299, 311, and 324 |
|
3) Author report “The role of the father’s educational level allows us to draw conclusions about his role in the sleep of his offspring. On the one hand, their involvement may be indirect, through excessive use of mobile devices, as it has been reported that an insecure parental attachment style leads to problematic use [45], and parental control also seems to have an impact on excessive use of these devices [46]”. Subsequently, authors add this argumentation “A study that looked at the characteristics of both parents for insufficient sleep during childhood found that the risk of poor sleep was higher when mothers (but not fathers) worked 35 hours or more per week, and the risk of children not getting enough sleep was higher when the father worked in construction and production [47]”. Could the authors explain how the two statements are connected? While the second statement seems understandable to me, I find it difficult to understand the relationship between the parent's attachment style, their level of education, and the quality of their children's sleep.
|
His observations in this regard make sense, so we eliminated two references. 1) Attachment has nothing to do with educational level, so this reference was eliminated and the information was expanded to connect the ideas. 2) We did not find more information from adolescents regarding the role of the father's educational level, so we thought that the information from children could give us some reference (given that adolescents still depend to some extent on parental control). One reference to children was eliminated and another one was left, explaining its connection with adolescents. 1) |
Lines 341-350 |
|
Minor comments: |
|
|
|
1) Please, put a double colon after "method" (row 14).
|
Double colon was placed after method
|
Line 14 |
|
2) In the keywords, I suggest reporting problematic Internet use first and then problematic mobile phone use (row 28).
|
The keyword was placed first: problematic Internet use and then problematic mobile phone use
|
Line 28 |
|
3) Why do the authors report the p-value on line 63 rather than the reference of the statement?
|
We appreciate the observation. Instead of the p value there should be the reference. Correction was made
|
Line 64 |
|
4) Did the authors use versions translated of instruments into the native language of Mexico? If so, could you please specify; if not, could you please describe the translation procedure. Then, it could be useful for reader to report some examples of items.
|
Clarification was made about the adaptation to Mexican Spanish for each instrument
A representative example of the items was given
|
Lines 151-156, 174-176
Lines140-141, 163-165 |
|
5) Descriptive characteristics of the sample could be reported in the sample paragraph rather than in the results.
|
Table 1 was summarized in the method section 2.1 and was therefore removed from the results.
|
Lines 109-114 |
|
Comments on the Quality of English Language I advise the authors to do a general check of the English even if overall it seems well written. |
It was reviewed |
|

Round 2
Reviewer 1 Report
Comments and Suggestions for Authors
All suggestions have been considered
Author Response
COMMENT: All suggestions have been considered
RESPONSE: We appreciate your suggestions.
Reviewer 2 Report
Comments and Suggestions for Authors
The following revision concerns round 2 of the manuscript "Problematic Use of Mobile Phones and Social Media on Sleep Quality of High School Students in Mexico City". Authors have improved the work based on the suggested comments. Minor suggestions are reported below.
Request of clarification
Could Table 2 be improved in visualization? For example, I suggest reporting p-value indices (*) for a couple of variables. In the specific case, looking at the table, for the difference between quality of sleep and “Participates in outdoor or recreational activities”, my question regards if differences are referred to YES/NO. In other words, on the “quality” of sleep the groups of people who answer “yes” are different from those who answer “no” to the statement “Participates in outdoor or recreational activities”. If this is correct, why are the significance indices positioned only near YES?
Suggestions
Author could replace “mobile phone” with “smartphone”.
Typos
Please, authors report title “Introduction” (line 30) in bold.
Please, authors check punctuation.
Author Response
COMMENTS 1: Request of clarification
Could Table 2 be improved in visualization? For example, I suggest reporting p-value indices (*) for a couple of variables. In the specific case, looking at the table, for the difference between quality of sleep and “Participates in outdoor or recreational activities”, my question regards if differences are referred to YES/NO. In other words, on the “quality” of sleep the groups of people who answer “yes” are different from those who answer “no” to the statement “Participates in outdoor or recreational activities”. If this is correct, why are the significance indices positioned only near YES?
RESPONSE 1: As suggested by reviewers we revised and clarified Table 2. We hope this facilitates interpretation of results.
COMMENTS 2: Suggestions
Author could replace “mobile phone” with “smartphone”.
RESPONSE 2: We thank you for this suggestion as it significantly improves the text. We made changes throughout the entire manuscript.
COMMENTS 3: Typos
Please, authors report title “Introduction” (line 30) in bold.
Please, authors check punctuation.
RESPONSE 3: We did and hoped we covered all typos. We added the edit and highlighted it in yellow. We also re read the text and made extensive edits that can be seen in track changes to make it more readable.